# Impact of Non-Covalent Interactions of Chiral Linked Systems in Solution on Photoinduced Electron Transfer Efficiency

**DOI:** 10.3390/ijms24119296

**Published:** 2023-05-26

**Authors:** Ilya M. Magin, Ivan A. Pushkin, Aleksandra A. Ageeva, Sofia O. Martianova, Nikolay E. Polyakov, Alexander B. Doktorov, Tatyana V. Leshina

**Affiliations:** 1Voevodsky Institute of Chemical Kinetics and Combustion, 630090 Novosibirsk, Russia; magin@kinetics.nsc.ru (I.M.M.); pushkin@kinetics.nsc.ru (I.A.P.); al.ageeva@gmail.com (A.A.A.); smartle636@gmail.com (S.O.M.); polyakov@kinetics.nsc.ru (N.E.P.); leshina@kinetics.nsc.ru (T.V.L.); 2Department of Natural Sciences, Novosibirsk State University, 630090 Novosibirsk, Russia

**Keywords:** non-covalent interaction, photoinduced electron transfer (PET), chiral associate, diastereomer, chemically induced dynamic nuclear polarization (CIDNP)

## Abstract

It is well-known that non-covalent interactions play an essential role in the functioning of biomolecules in living organisms. The significant attention of researchers is focused on the mechanisms of associates formation and the role of the chiral configuration of proteins, peptides, and amino acids in the association. We have recently demonstrated the unique sensitivity of chemically induced dynamic nuclear polarization (CIDNP) formed in photoinduced electron transfer (PET) in chiral donor–acceptor dyads to non-covalent interactions of its diastereomers in solutions. The present study further develops the approach for quantitatively analyzing the factors that determine the association by examples of dimerization of the diastereomers with the RS, SR, and SS optical configurations. It has been shown that, under the UV irradiation of dyads, CIDNP is formed in associates, namely, homodimers (SS-SS), (SR-SR), and heterodimers (SS-SR) of diastereomers. In particular, the efficiency of PET in homo-, heterodimers, and monomers of dyads completely determines the forms of dependences of the CIDNP enhancement coefficient ratio of SS and RS, SR configurations on the ratio of diastereomer concentrations. We expect that the use of such a correlation can be useful in identifying small-sized associates in peptides, which is still a problem.

## 1. Introduction

The abilities of natural peptides to self-organize into various supramolecular assemblies—dimers, oligomers, fibrils, and other nanostructures—are currently of great interest to scientists in multiple specialties [1,2,3,4]. Despite numerous studies, the molecular mechanisms of such structures’ formation are still unknown. In addition, it is known that the optical orientation (D/L) of amino acids in proteins and peptides plays an important role: changes in the chiral orientation lead to changes in the structures and bioactivity of peptide assemblies [1,2]. It is associated with violations of the folding processes in a number of proteins [5,6,7,8]. The study of the role of the optical configuration of amino acids in folding disturbance is an urgent problem because several acute diseases are associated with these processes [6,8,9,10]. Such disorders are currently considered as one of the leading causes of neurodegenerative diseases, including the curse of modernity—Alzheimer’s disease. Instead of the normally present L-isomer, the appearance of D-amino acids in several proteins and peptides in the brain leads to the formation of highly disordered structures that cannot be studied using high-resolution NMR and X-ray techniques [6]. This problem is proposed to be solved using small model systems for studying the features of the molecular dynamics of systems with L- and D-amino acids by in vitro and in silico methods [6]. An example of a successful combination of these methods would be the work [9].

Earlier, we carried out a modification of the above approach, which made it possible to study both problems: establishing the nature of differences in the reactivity of small chiral systems with different optical configurations and their tendency to associate [11]. In a modified version, in order to trace the differences between the reactivity of optical isomers, intramolecular photoinduced electron transfer (PET) in chiral linked systems — dyads was used [11,12,13]. PET occurred in dyads where the L- or D-residues of tryptophan (Trp) or N-methylpyrrolidine served as donors, while non-steroidal anti-inflammatory drugs (NSAIDs) (R/S) naproxen and (R/S) ketoprofen were acceptors (Figure 1).

It should be noted that ET and its photoinduced variant (PET) are universal processes used to model elementary stages in biological systems [11,12,13,14].

The joint application of chemically induced dynamic nuclear polarization (CIDNP), fluorescence quenching techniques, and molecular modeling have demonstrated a difference in the structures of diastereomers with the different optical configurations of donors and acceptors and in the efficiency of PET in these dyads. CIDNP effects formed during the UV irradiation of donor–acceptor dyads in solutions demonstrated a high sensitivity to the intramolecular ET and intermolecular interaction [11,12,13]. For these dyads, CIDNP is formed in a biradical zwitterion (BZ) obtained via intramolecular ET (Figure 1), while intermolecular interaction, also affecting CIDNP, includes the formation of hydrogen bonds between two diastereomers.

The impact of hydrogen bonds, as an example of non-covalent interaction, on the PET efficiency was also demonstrated in [14].

The phenomenon of CIDNP is the appearance of signals with a non-Boltzmann population of nuclear spin sublevels (enhanced absorption or emission) in the NMR spectra of products obtained from the radical precursor during the reaction in the NMR spectrometer probe. In our case, the radical precursor is BZ. An analysis of the CIDNP effects allows one to obtain a portrait of the radical precursors of polarized products (hyperfine interaction constants (HFI), g-factors, etc.) [15]. Today, CIDNP is one of the most informative, albeit indirect, methods for identifying short-lived radical particles [15]. Two peculiarities of CIDNP effects in PET in chiral linked systems have been established [11,12,13]. First is the difference in CIDNP enhancement coefficients for RS and SS diastereomers. This difference in the enhancement coefficients of CIDNP formed in the act of back electron transfer in BZ was named spin selectivity (hereinafter referred to as **K**) [12]. The spin selectivity for the diastereomers of dyads I and II is about 2. It was suggested that the variation in the distribution of spin density in BZ with different optical configurations is most likely due to the difference in the geometry of the BZ of RS and SS diastereomers [12].

The second peculiarity of the CIDNP in the chiral linked systems is the dependence of **K** on the diastereomers’ concentration ratios upon their joint UV irradiation in solutions. Since the values of the CIDNP absolute enhancement coefficients refer to a pair of radicals (or BZ in this case), they and their ratios should not depend on the concentrations of other diastereomers. Thus, this dependence suggests some intermolecular interaction or bimolecular chemical reaction between the diastereomers in the solution.

For dyads I and II containing NPX, ET was suggested to occur in dimers of dyads [12]. X-ray data on the diastereomers of the dyad II provided direct evidence for the formation of dimers. In the case of dyad I, the indirect evidence of dimerization was demonstrated by comparing the experimental (**K**) and calculated (**K_c_**) values on the ratios of the concentrations of diastereomers under UV irradiation of their mixtures. When calculating the dependence of **K**_c_ on the ratio of the concentrations of diastereomers, the well-known Frank hypothesis describing chiral enrichment was used [12,16]. Applying the Frank principle to describe the dimerization of chiral diastereomers using the dependence of the **K**_c_ on the ratios of their concentrations was performed under the following assumptions: Chiral particles will tend to associate with partners of the same optical orientation, creating homodimers (RS-RS, SS-SS) and preventing the formation of heterodimers (RS-SS) [12,13]. The dependence of the **K**_c_ value on the ratio of diastereomer concentrations, calculated under the assumption of the complete dominance of homodimers in solution, is shown in Figure 2. The experimental **K** values show the same dependence.

In addition, dimers have been assumed to be formed with a high dimerization equilibrium constant of the order of 10^5^ M^−1^ and with a tenfold predominance of the homo configuration.

As for dyad III, the conclusion about the possibility of dimer formation has been made based on the analysis of only the experimental dependence for the ratio of observed CIDNP enhancement coefficients on the ratios of the concentrations of diastereomers under UV irradiation of their mixtures [11,17,18,19].

In this paper, using a CIDNP formed in PET in dyads II and III and a modified calculating approach, we try to trace the impact of homo-, heterodimers, and monomers, as well as the structure of dyads in the dimerization process. The modification of the approach includes numerically solving a differential equations system, to determine the stationary concentrations of diastereomers in the form of dimers. In addition, the values of the dimerization equilibrium constants and the probabilities of the formation of homo- and heterodimers vary over a wide range, including taking into account the possible contribution of monomers. The dependences of **K** for the dyads II and III diastereomers on the ratio of their concentrations are measured and compared with the calculated analogues of these dependences. The peculiarities of the chemical polarization effects in associated linked systems are also explored using the data of the time-resolved CIDNP. We hope it will allow us to get closer to understanding the roles of homo- and heterodimers in the formation of small associates in the chiral systems.

## 2. Results and Discussion

The present article is devoted to studying the influence of non-covalent intermolecular interactions on CIDNP in dyads II and III diastereomers. Previously, it was demonstrated by fluorescence and CIDNP methods that the CIDNP in dyads II and III is formed in the BZ resulting from the quenching of the excited states of chromophores (see Figure 1) [8,11,12,13]. In the case of dyad II, the chromophore is naproxen (NPX), while for dyad III, it is tryptophan (Trp).

An analysis of the time-resolved CIDNP effects in the photoinduced ET in the dyad II, performed in this work, according to existing rules, confirmed the realization of ET through the formation of BZ [15]. In addition, CIDNP effects observed under the UV irradiation of a mixture of diastereomers in solutions have shown a strong dependence of the **K** values of diastereomers on the ratio of their concentrations for dyads I–III (Figure 3).

To explain such different dependences shown in Figure 3, a new approach describing the process of dyad association has been developed and tested.

### 2.1. Description of a Modified Approach to Calculate CIDNP Dependence on the Ratio of Diastereomer Concentrations

In the previous work, calculations of the dependences of the ratio of the CIDNP coefficients for diastereomers on the ratios of their concentrations were carried out under the assumption of their dimerization. It was assumed that diastereomers were predominantly found in the composition of homodimers [12,16]. The approximation of quasi-stationary concentrations was used in the calculation. In this work, the concentrations of dimers and monomers were obtained by numerically solving the kinetic equations for the dimerization reaction. The assumption about the formation of polarization with different efficiencies in different forms of diastereomers is shown in Figure 2. α, β, and γ are the efficiencies of polarization formation in homo- and heterodimers, as well as in monomers, respectively.

The kinetic Equation (1) describing the dimerization was solved numerically and the values of the stationary concentrations were used in Equation (2) for the ratio of the CIDNP enhancement coefficient **K** of diastereomers.

(1)
$$\begin{array}{l} \frac{d[SS]}{dt}=-2\cdot K_{SS,SS}^{+}\cdot {\left[SS\right]}^{2}+K_{SS,SS}^{-}\cdot [D_{SS,SS}]-K_{SR,SR}^{+}\cdot [SS]\cdot [SR]+K_{SS,SR}^{-}\cdot [D_{SS,SR}]\\ \frac{d[SR]}{dt}=-2\cdot K_{SR,SR}^{+}\cdot {\left[SR\right]}^{2}+K_{SR,SR}^{-}\cdot [D_{SR,SR}]-K_{SS,SR}^{+}\cdot [SS]\cdot [SR]+K_{SS,SR}^{-}\cdot [D_{SS,SR}]\\ \frac{d[D_{SS,SS}]}{dt}=K_{SS,SS}^{+}\cdot {\left[SS\right]}^{2}-K_{SS,SS}^{-}\cdot [D_{SS,SR}]\\ \frac{d[D_{SS,SS}]}{dt}=K_{SR,SR}^{+}\cdot {\left[SS\right]}^{2}-K_{SS,SS}^{-}\cdot [D_{SS,SR}]\\ \frac{d[D_{SR,SR}]}{dt}=K_{SS,SS}^{+}\cdot {\left[SS\right]}^{2}-K_{SR,SR}^{-}\cdot [D_{SR,SR}]\\ \frac{d[D_{SS,SR}]}{dt}=K_{SS,SR}^{+}\cdot [SS][SR]-K_{SS,SR}^{-}\cdot [D_{SS,SR}] \end{array}$$


Here, K^+^ and K^−^ are the dimer (D) formation and dissociation constants.

(2)
$${\tilde{K}}_{c}=\frac{\alpha_{SR}[D_{SR,SR}]+\beta_{SR}\frac{\left[D_{SS,SR}\right]}{2}+\gamma [SR]}{\left[D_{SR,SR}]+\frac{\left[D_{SS,SR}\right]}{2}+[SR\right]}\cdot \frac{\left[D_{SS,SS}]+\frac{\left[D_{SS,SR}\right]}{2}+[SS\right]}{\alpha_{SS}[D_{SS,SS}]+\beta_{SS}\frac{\left[D_{SS,SR}\right]}{2}+\gamma [SS]}$$


For a correct comparison with the observed ratio of polarization intensities, we should normalize the **K_c_** ratio to the total concentration of diastereomers [RS/SS]_0_ and its biradical zwitterion’s quantum yields φ. The result is described by Formula (3).

(3)
$$K_{c}=\frac{\alpha_{SR}[D_{SR,SR}]+\beta_{SR}\frac{\left[D_{SS,SR}\right]}{2}+\gamma [SR]}{2[D_{SR,SR}]+[D_{SS,SR}]+[SR]}\cdot \frac{2[D_{SS,SS}]+[D_{SS,SR}]+[SS]}{\alpha_{SS}[D_{SS,SS}]+\beta_{SS}\frac{\left[D_{SS,SR}\right]}{2}+\gamma [SS]}\cdot \frac{\varphi_{SS}}{\varphi_{SR}}$$


The modified approach was tested on the results of experiments in the dyad I presented in [12]. The experimental concentration dependence was simulated under the condition of high dimerization equilibrium constants and the prevalence of the contribution to CIDNP from homodimers compared to heterodimers (α >> β). The parameters used in [12] and substituted into Equation (3) give a result completely similar to that shown in Figure 2 (see Appendix A).

Note that considering the contributions to the total CIDNP from monomers, which is possible within the framework of the new approach, does not significantly affect the calculation result. The examples of the influence of the dimerization equilibrium constants variation and CIDNP coefficients of monomers on the **K_c_** value are presented in Appendix A.

### 2.2. Dependences between the K and the Ratio of Dyad II Diastereomer Concentrations: Comparison of Experimental and Calculated Results

The diastereomer’s CIDNP coefficients were measured under the conditions of the equal total concentrations of diastereomers in the solution with a change in their relative concentrations. Thus, considering the same extinction coefficients of the diastereomers, the sample’s optical properties, i.e., the amount of absorbed light, were preserved. The dependence of the ratio of the CIDNP coefficients of diastereomers for dyad II differs significantly from that of dyad I described above. In addition to measuring the enhancement coefficients for the aromatic protons of dyad II from the pseudo-stationary CIDNP (PSS) data, we also measured the CIDNP time dependence using the TR-CIDNP technique (see the Experimental Part). The time dependence of CIDNP is shown in Figure 4.

Figure 5 shows the experimental dependence of **K** on the ratio of diastereomer concentrations. It includes the **K** values measured using the PSS and TR CIDNP techniques. The CIDNP intensities measured at zero time delay were used in the latter case.

To describe the dependence obtained, in contrast to the previous case, we must abandon the assumption that the formation of polarization in homodimers is much more efficient than in heterodimers (α >> β). The ratio of α_SR_ to α_SS_ is determined from the experiment, so the experimental dependence can be fitted with a set of parameters α_SR_ = 1.8, α_SS_ = 1, β_SR_ = 1.6, and β_SS_ = 5.5. The values of γ for both diastereomers are equal to 0, which, at dimerization equilibrium constants similar to those for the dyad 1–2 × 10^5^ M^−1^, has little effect on the result. The results of varying the dimerization equilibrium constants of dyad II, as well as the parameters β and γ, are presented in Appendix A.

### 2.3. Dependences between the K Values and the Ratio of Dyad III’s Diastereomer Concentrations: Comparison of Experimental and Calculated Dependences

The study of the dyad III by CIDNP and fluorescence quenching methods has already been performed, and it demonstrated several peculiarities [14,15]. The ratio of the observed CIDNP coefficients of the SS and (RS, SR) configurations is 9.8, instead of 2, detected for dyads I and II. The RS and SR optical configurations show the same observed CIDNP coefficients. In addition, the CIDNP spectrum contains additional polarized lines with the same intensity ratios as observed in the SS and RS, SR configurations. Polarized signals apparently belonging to the products can be seen near the methylene protons of the tryptophan moiety (3.2 ppm, Figure 6).

The dependences of the diastereomers’ CIDNP coefficient ratios on the ratios of their concentrations were also studied for this system earlier [8,16]. However, the dependencies for the CIDNP coefficients have been analyzed only qualitatively, since neither the ratio of BZ concentrations for diastereomers nor **K** values were defined. In turn, to calculate the dependence of **K** on the concentration ratio within the framework of the developed approach, it is necessary to determine the absolute CIDNP coefficients, based on the knowledge of the BZ concentration. Meanwhile, due the product’s contribution to the intensity of equilibrium signals, these options cannot be obtained using the ratio of the intensities of the polarized and equilibrium NMR signals. That is why we tried to use a different way to estimate the absolute **K** values for the diastereomers of the dyad III. Since the formation of products under the UV irradiation of these diastereomers does not allow the use of equilibrium NMR signals, we proposed to analyze the relationship between the intensity of polarized signals and the fraction of light absorbed by each diastereomer in the mixture. Using the Bouguer–Lambert–Beer (BLB) law, this relationship can be represented as 
$\left(1-10^{-ELI_{pol}}\right)$
. *E*, *L*, and *C* are the extinction coefficient, optical path length, and concentration. It is assumed that proportionality is preserved between the BLB dependences for the dyad in the ground and excited states, namely, 
$\left(1-10^{-ELC}\right)$
~
$\left(1-10^{-ELI_{pol}}\right)$
. This is proportionality and not equality because, as mentioned above, ET is not the only process that occurs under the UV irradiation of dyad III. At the same time, the synchronous behavior of the CIDNP effects of the protons of the initial dyad and products allows us to assume that the latter is also formed from BZ and, accordingly, depends in the same way on the fraction of absorbed light (Figure 7).

Thus, it can be assumed that the different fitting parameters E_1_ and E_2_ appear due to differences in the BZ concentrations for the RS and SS diastereomers. Then, 
$1-10^{{-E_{1}LC}_{BZ}^{SS}}=1-10^{{-E_{2}LC}_{BZ}^{SR}}$
, and 
$E_{1}C_{BZ}^{SS}L=E_{2}C_{BZ}^{SR}L$
. From the fitting results, the values E_1_ = 0.178 and E_2_ = 0.953. Then, the ratio BZ(SS)/BZ(SR) = E_2_/E_1_ = 5.4. If we divide the observed coefficient by the difference in the BZ concentrations of the SS and SR diastereomers, we obtain the **K** value of 1.8, which is close to those for dyads I and II [12].

Now, using the obtained values of **K**, it is possible to compare the experimental dependence of **K** on the ratio of diastereomer concentrations with the calculated one (Figure 8).

### 2.4. Influence of the Homo-, Heterodimer, and Monomer Contributions and Magnitudes of Dimerization Equilibrium Constants on the CIDNP Efficiency in ET

Thus, the new approach allows us to describe the dependences of the **K** on the ratio of the diastereomer concentrations for dyads I–III. This became possible due to the using of the experimental **K** values and the variation of the ET probabilities in homo- and heterodimers according to Figure 2. The developed approach also makes it possible to take into account the participation of monomers (γ). Thus, the NMR signals of the polarized protons of the RS and SS diastereomers of the I–III dyads can contain contributions from all types of particles from Figure 2. At the same time, the contributions of homo-, heterodimers, and monomers, as well as the dimerization equilibrium constants, change depending on the ratio of the diastereomer concentrations under the UV irradiation of their mixtures in solution.

Parameters for fitting the **K** dependences on the ratio of the dyads I–III diastereomer concentrations are presented in Table 1. Therefore, the experimental dependence for dyad I coincides with the calculated one at a tenfold excess of the contribution to CIDNP from homodimers (α) compared to heterodimers (β), and using a high dimerization equilibrium constant. In the case of dyad II, on the contrary, to coincide the calculation with the experiment, it is necessary to suggest that ET occurs in heterodimers which make the main contribution to CIDNP (Table 1 and Figure 5). It is also worth noting that, for dyad II, in which the dependence of the CIDNP intensity on the RS/SS ratio is measured, the best agreement with the calculation is obtained when the predominant contribution is made by the β dimer (SS-SR), in which the SS isomer absorbs light and creates CIDNP with a smaller coefficient.

As for dyad III, for which the agreement between the experimental dependence of **K** on the SS/SR ratio and the calculated one is worse than for dyads I and II, there are several reasons for this. This is a substantial difference in the SS and RS, SR optical configuration properties. In particular, it is known that the quenching of the Trp singlet excited state via ET is not the only way for dyad III, and it seems that it is not the main one for the RS and SR configurations [17]. This can result from differences in the spatial structure of the diastereomer’s BZ since the SS and SR isomers are the same molecules differing only in the optical configuration of the tryptophan fragment. The combination of two factors—the small CIDNP coefficient of the SR configuration compared to SS and the prevalence of SS-SR heterodimers in the reaction mixture—leads to the observed dependence of **K** on the SS/SR ratio. Therefore, in the experiment, we observe a sharp increase in the ratio of the CIDNP SS/SR in the initial part of the curve, due to differences in the efficiency of ET, and then a decrease due to the contribution of dimers with minor CIDNP effects. In addition, the substantial difference in the efficiency of the ET in the SS and SR diastereomers of dyad III makes this dyad less suitable for the application of the chosen approach. It is worth noting that the ET contribution to the NPX excited state quenching for the dyads I and II diastereomers is the main one and their BZ concentrations are close to each other. Therefore, the calculation results cannot reproduce the initial **K** growth in the case of dyad III.

Now, we should discuss the influence of both partners in the dyads on the ongoing processes. It follows from the data in Table 1 that the role of heterodimers in the case of dyad I is significantly different from that of dyads II and III. In both dyads with tryptophan, the probability of the heterodimer formation is comparable or greater than that of homodimers. The differences between II and III, in turn, indicate the influence of the second partner on the probability of heterodimer formation. Therefore, the results of molecular modeling indicate the preferential arrangement of the tryptophan fragment in dimers opposite to the NPX or KP residues: the head-to-tail arrangement [17]. This suggests that the ET in dimers can occur according to Figure 2 as well as between two diastereoisomers, when the excited state of tryptophan in one diastereomer is quenched by another diastereomer in the ground state.

These processes are shown in Figure 3 which involves intra- and intermolecular ET and degenerate electron exchange (EE) between the BZ and the dyad in the ground state.

Figure 3 presents intra- and intermolecular ET occurring within the homodimer, but, for heterodimers, there should be a similar scheme.

The hypothesis about the possibility of two types of ET, mono and bimolecular, as well as about degenerate electronic exchange, arose from the analysis of the shape of the kinetic curve of the time-resolved CIDNP of dyad II protons (Figure 4). The observation of the increase of CIDNP intensity up to 20 µs indicates the participation of not only geminate but also homogeneous stages in CIDNP formation. In this case, both processes lead to the appearance of a CIDNP of the same sign. Since the sign of the CIDNP corresponds to the quenching of the singlet excited state, both in the mono and bimolecular process, it is evident that back ET occurs in the BZ and radical ion pair (RIP) – analog BZ in the dimer (left side of the Figure 3). In this case, the observed shape of the kinetic curve can be a superposition of the CIDNP effects from radical pairs with different lifetimes. Here, the lifetime is likely to be longer for RIP due to the probable spatial difficulties in back ET. Note that long-lived charge transfer stages correspond to a reversible ET in a linked system [20].

Thus, the presence of weak non-covalent interactions between diastereomers shown in Figure 3 also affects the course of the CIDNP time dependence. It is worth noting that picosecond absorption spectroscopy studies of “tyrosine–tryptophan dyads”, which are important structural motifs in a number of enzymes, have demonstrated a significant effect of weak non-covalent interactions on the efficiency of consecutive electron–proton transfer (PCET) [21].

In addition, the appearance of various forms of dependences, including cases where heterodimers (β) predominate over homodimers (α), allows us to assume that Frank’s principle is not always valid for small molecules forming small associates. At the same time, it is known that the formation of large assemblies, such as chiral polymers, as well as the products of chiral enrichment in the Soai reaction is known to follow the Frank principle: the chiral optical isomer catalyzes its own reproduction and reduces the probability of another isomer appearance [12,22,23]. In our case, this means that the formation of homodimers should prevail over heteroanalogues, but, as this work shows, this is not always the case. Since the mechanisms of small associate formation are less studied than for large ensembles, the establishment of these mechanisms may be useful for understanding the role of weak non-covalent interactions in the association of biomolecules [1].

## 3. Materials and Methods

The dyad II and dyad III were synthesized using a method described earlier [18,24]. Solutions of KP-Trp dyads were prepared in deuteroacetonitrile (99.8% D, Aldrich, St. Louis, MO, USA) for CIDNP measurements. Solutions of NPX-Trp dyads were prepared in deuteroacetonitrile/deuterobenzene (CIL, 99.4% D) mixture in proportion of 60/40.

The concentration dependencies of diastereomers were performed using a total concentration of 5 × 10^−3^ M for dyad II and 8 × 10^−3^ M for dyad III, increasing the first diastereomer concentration and decreasing the second.

^1^H NMR and photo-CIDNP experiments were provided on a DPX-200 NMR spectrometer (Bruker, Germany, 200 MHz 1H operating frequency, P(π/2) = 2.5 µs). A Lambda Physik EMG 101 MSC eximer laser was used as a light source (308 nm, 100 mJ at output window, 20 mJ/pulse in sample volume pulse duration 15 ns) in the CIDNP experiments. The samples in standard 5 mm Pyrex NMR tubes were irradiated directly in the NMR probe of DPX-200 NMR spectrometer. The samples were bubbled with argon for 15 min to remove dissolved oxygen just before photolysis.

The photo-CIDNP was performed as time-resolved (TR) and pseudo-steady-state (PSS) experiments. To carry out PSS experiments, a standard pulse sequence was used: presaturation–delay1–pulse τ(π)–delay2 (16 laser flashes with repetition rate 50 Hz during delay2)–observation pulse τ(π/2)–acquisition. Delay1/delay2 ≈ 1.1 was used to remove residual signals of solvents and solutes. After laser irradiation, the ^1^H NMR spectra of polarized products were recorded. This allows a CIDNP spectrum to be obtained with a much better signal-to-noise ratio, albeit with a loss of time resolution.

## 4. Conclusions

Thus, the high sensitivity of the CIDNP effects arising under the UV irradiation of a mixture of donor–acceptor dyad diastereomers in solutions to weak non-covalent interactions between them has been demonstrated. The developed approach makes it possible, by comparing the calculated and experimental CIDNP coefficients, to establish a relationship between the efficiency of PET and the proportions of associates (homo- and heterodimers), as well as monomers present in solutions. The CIDNP signal in this case is a superposition of contributions predominantly from homo- and heterodimers. A comparison of the dependence of K on the ratio of their concentrations with the results of the calculations also showed the role of the dimerization equilibrium constants. A study using TR CIDNP allowed us to propose a scheme detailing the peculiarities of ET in dimers and confirms the diastereomers association. In addition, the authors believe that the use of CIDNP methods provides additional opportunities for studying the elementary stages of the small linked systems association and identifying the influence of chiral centers on these processes.

## Data Availability

Not applicable.

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
