# Peer review of "Impact of Non-Covalent Interactions of Chiral Linked Systems in Solution on Photoinduced Electron Transfer Efficiency"

_ijms, 2023, doi:10.3390/ijms24119296_

Round 1

Reviewer 1 Report

Comments and Suggestions for Authors

Manuscript Number: ijms-2397602

entitled: Non-covalent interactions between chiral linked systems in solutions studied using photoinduced electron transfer

This is an interesting scientific study. Therefore, the manuscript is suitable for publication in the International Journal of Molecular Sciences after considering the below comments:

I'm afraid I have to disagree with the author's thesis: page 2 “Today, CIDNP is one of the most informative, albeit indirect, methods for studying radical processes [15].” Since the early works of Gomberg, a lot has been discovered, although there were no modern analytical methods such as EPR or NMR.

In the range 1968 – 2023 Scopus CIDNP, find 1211 documents only. (please see attached file). The documents are mainly from the USA, Russian Federation, and Germany.

In the Conclusion part, please indicate the advantages of the CIDNP technique over others for a broadly understood reader.

Author Response

Response to reviewer 2

Comment  1. I'm afraid I have to disagree with the author's thesis: page 2 “Today, CIDNP is one of the most informative, albeit indirect, methods for studying radical processes [15].” Since the early works of Gomberg, a lot has been discovered, although there were no modern analytical methods such as EPR or NMR.

In the range 1968 – 2023 Scopus CIDNP, find 1211 documents only. (please see attached file). The documents are mainly from the USA, Russian Federation, and Germany.

 Answer 1. Authors can only assume that when referring to Gomberg, the reviewer means the study of stable radicals. Here, undoubtedly, EPR and NMR (Rowan, Hahn, Mims, Webb, Tsvetkov, Sagdeev, Molin) made the main contribution. But for short-lived (nanosecond range) free radicals this is not the case at all. To eliminate discrepancies, the authors have changed the phrase that caused this comment: “To date, CIDNP is one of the most informative, albeit indirect, methods for identifying short-lived radical particles [15]”.  Note that the observation of polarized NMR signals of the products unambiguously indicates their radical origin, in contrast, for example, to the data of EPR spectroscopy.

Comment  2. In the Conclusion part, please indicate the advantages of the CIDNP technique over others for a broadly understood reader.

Answer 2. The conclusion has been rewritten, taking into account this recommendation: “The authors believe that the use of CIDNP methods provides additional opportunities for studying the elementary stages of the small linked systems association and identifying the influence of chiral centers on these processes.”

Reviewer 2 Report

Comments and Suggestions for Authors

The manuscript "Non-covalent interactions between chiral linked systems in solutions studied using photoinduced electron transfer" is a great contribution in the field of self-assembled chiral configuration of proteins photoinduced by the electron transfer. The novelty of the manuscript consists in highlighting the factors involved in the association process by means of dimerization of dyads with RS, SR and SS configurations.

The authors also highlighted the mechanism of formation of small associates through non-covalent interactions by determination of equilibrium constants and the probability of homo-/ heterodimers formation based on the contribution of the monomers. They also used time-resolved CIDNP to evaluate the effects of the chemical polarization in associates.

Overall, the paper stands out through theoretical and experimental approaches well performed and demonstrated, so that it can be accepted for publication in the present form.

Author Response

Comment

The manuscript "Non-covalent interactions between chiral linked systems in solutions studied using photoinduced electron transfer" is a great contribution in the field of self-assembled chiral configuration of proteins photoinduced by the electron transfer. The novelty of the manuscript consists in highlighting the factors involved in the association process by means of dimerization of dyads with RS, SR and SS configurations.

The authors also highlighted the mechanism of formation of small associates through non-covalent interactions by determination of equilibrium constants and the probability of homo-/ heterodimers formation based on the contribution of the monomers. They also used time-resolved CIDNP to evaluate the effects of the chemical polarization in associates.

Overall, the paper stands out through theoretical and experimental approaches well performed and demonstrated, so that it can be accepted for publication in the present form.

Answer Thank a lot for the comments

Reviewer 3 Report

Comments and Suggestions for Authors

In this paper, the authors report a system of electron transferring induced by photo. The work is of theoretical meanings and merit publication. However, major revisions are needed.

(1)    Abstract is too long and need to be made concise. The novelty and significances should be emphasized in abstract;

(2)    Key word “diastereomers” should be “diastereomer”;

(3)    The character size of the figures should be adjusted. Some are too large.

(4)    In experimental part, the solvent for NMR tests should be given.

(5)    In mechanism discussion, the authors proposed that radicals are involved. This should be supported by references, e.g. 10.1016/j.cclet.2023.108489

Comments on the Quality of English Language

fine

Author Response

Comment 1  Abstract is too long and need to be made concise. The novelty and significances should be emphasized in abstract;

Response 1 The size of Abstract does not exceed the size recommended by the IJMS rules. The novelty and significance are described in details in the Introduction.

Comment 2 Keyword “diastereomers” should be “diastereomer”;

Response 2  It was corrected.

Comment 3 The character size of the figures should be adjusted. Some are too large.

Response 3 Figures were corrected.

Comment 4 In the experimental part, the solvent for NMR tests should be given.

Response 4 It was corrected.

Comment 5 In mechanism discussion, the authors proposed that radicals are involved. This should be supported by references, e.g.10.1016/j.cclet.2023.108489

Response 5 The article 10.1016/j.cclet.2023.108489 is published in Chinese and not available in Russia. Anyway, the formation of radicals in the systems under study via photoinduced electron transfer was proved in this paper as well as in our previous publications by detection of CIDNP effects.

Round 2

Reviewer 3 Report

Comments and Suggestions for Authors

accept